# Natural Deep Eutectic Solvents in the Synthesis of Inorganic Nanoparticles

**DOI:** 10.3390/ma16020627

**Published:** 2023-01-09

**Authors:** Olga Długosz

**Affiliations:** Faculty of Chemical Engineering and Technology, Cracow University of Technology, 31-155 Cracow, Poland; olga.dlugosz@pk.edu.pl

**Keywords:** natural deep eutectic solvents, NDES, metal nanoparticles, metal oxide, inorganic nanoparticles

## Abstract

Natural deep eutectic solvents (NDESs), as a new type of green solvent, are used in many fields, including industry in extraction processes, medicine, pharmaceuticals, metallurgy, electrodeposition, separations, gas capture, biocatalysis and nanotechnology. Mainly due to their properties, such as simple preparation, environmental friendliness, biocompatibility and multifunctionality, they are being used in various fields of industry. This review aims to provide insight into the applications of natural deep eutectic solvents, specifically in nanotechnology processes. It focuses on the description of NDES and how their physicochemical properties are used to obtain functional nanomaterials, including metals, metal oxides and salts. It highlights how the use of NDESs to obtain a wide range of inorganic nanoparticles enables the elimination of disadvantages of traditional methods of obtaining them, including reducing energy consumption and functionalising nanoparticles in situ. In conclusion, recent advances and future directions in the development and applications of NDESs in nanotechnology are discussed with the aim of identifying unexplained scientific questions that can be investigated in the future.

## 1. Introduction

In recent years, technological developments have overcome many problems and the rate of evolution of new types of materials and composites confirms that it is possible to develop products dedicated to specific applications. Some of the interesting materials that have been gaining popularity in the last 10 years are deep eutectic solvents (DESs) [1,2]. The number of scientific papers on these materials increases every year, and there are more and more areas in which natural deep eutectic solvents (NDESs) are used [3]. The uses of NDESs in electronics, medicine and biochemical processes are the most obvious and best described in the literature [4]. Nanotechnology may be added to the research areas in which NDESs are becoming increasingly important as solvents, reaction media, stabilising materials, nanoparticle modifiers and many more [5].

The term deep eutectic solvent was introduced relatively recently, namely, in 2003 [6] (Figure 1). As a new class of solvents, these solutions quickly began to gain popularity and, just five years later, the first paper on the use of DESs for nanoparticle synthesis was published [7]. Observing the potential for DESs, research was started to find combinations of components that would enable the production of more environmentally friendly solvents that would continue to offer the advantages of DESs. As a result of this research, a group of NDES comprising substances found in nature, specifically in primary metabolites, was identified in 2011 [8]. In the following years, new types of DESs were also developed, i.e., supramolecular deep eutectic solvents (SUPRADES) with cyclic oligosaccharides as acceptors of hydrogen bonds (HBAs) or hydrophobic deep eutectic solvents (HDESs) composed of hydrophobic compounds, such as tetrabutylammonium bromide, menthol, thymol and fatty acids as HBAs, along with long-chain alcohols and carboxylic acids as donors of hydrogen bonds (HBDs) [9].

NDESs, which can be composed of several compounds, allow for the preparation of nanoparticles with well-defined sizes and shapes [10,11]. When using them in the synthesis of nanomaterials, NDES can play the role of redox agent, stabiliser, supramolecular template, reaction environment or pH regulator, all with no need to introduce additional reactants. Selecting the composition of an NDES influences the viscosity, polarity, surface tension, hydrogen bonding and surface characteristics of nanomaterials, which directly affects the mass and energy transport properties of nanostructures [12]. Furthermore, DES components can modify nucleation and growth mechanisms by neutralising charges and passivating individual crystal surfaces, which dictates the growth along preferred crystallographic directions. Combining their great properties and broad perspectives, it is feasible to develop advanced nanostructures in an anhydrous medium [13,14].

However, before achieving this, it is necessary to first solve several challenges posed to researchers dealing with natural deep eutectic solvents. To make NDESs useful and applicable on a large scale, it is necessary to develop a universal nomenclature, develop low-viscosity NDESs, design methods for their preparation that take increased scale into account and develop NDESs that are insoluble in water [15]. The research on pro-ecological methods of obtaining NDESs at an increased scale is important for economic, ecological and technological reasons. Without the development of efficient methods for the synthesis of NDESs and the possibilities for their large-scale use, among others, the preparation of nanomaterials is limited.

This review presents what impact NDESs have on nanotechnology and how their use in nanomaterials may affect nanotechnology in the future. The development of DESs and NDESs has been analysed since 2011, which is when the concept of natural deep eutectic solvents was first introduced in nanotechnology. In particular, the possibility of obtaining inorganic nanoparticles using NDESs is highlighted, along with the relevance that the properties of NDESs may have on the features and applicability of nanomaterials [16].

## 2. Definition of NDES and Their Classification

### 2.1. Eutectic Mixtures

Similarly, to ionic liquids (IL), eutectic mixtures are classified as neoteric solvents. A eutectic mixture is “an approximately reversible, isothermal, non-reactive mixture of different components during cooling of a liquid system, resulting in a lower melting point of the system compared to the melting points of pure compounds” [17]. Such mixtures can consist of two components but can also be multi-component mixtures. However, the melting point of the mixture, which is described by determining the eutectic point (T_eut_), remains their most important property. At the eutectic point, the mixture reaches a minimum melting point. Below T_eut_, the entire mixture solidifies. The difference in the freezing point (T_f_) in the eutectic composition of a binary mixture compared with the freezing point of a theoretical ideal mixture, i.e., ΔT_f_, is related to the magnitude of the interaction between the components (Figure 2). The greater the interaction is, the greater ΔT_f_ will be [16]. Ionic liquids include liquid electrolytes, ionic alloys, molten and liquids salts, and ionic glasses. In contrast, deep eutectic solvents are composed of compounds in which the main interactions that occur between the components are hydrogen bonds. In addition, DESs can be biodegradable and less hazardous, which, when combined with their lower price, makes them an attractive alternative to IL [18].

### 2.2. Comparison of Deep Eutectic and Ionic Liquid Mixtures

Deep eutectic solvents belong to the group of eutectic mixtures. Due to their unique properties, DESs are a new class of sustainable solvents that are finding increasing applications in green solvent engineering. They are compared closely with ionic liquids but this is a separate group of solvents. In contrast to ionic liquids, DESs do not consist of ions. Among the advantages of DESs replacing ionic liquids are the fact that they are easier to synthesise, are less expensive, have no by-products and do not require purification [12]. In addition, due to their composition, they are often biodegradable and non-toxic. Common characteristics of DESs and IL include high thermal stability, low volatility, low vapour pressure, high thermostability and polar nature.

Deep eutectic solvents are divided into several types [19]. The first group of DESs (type I) includes mixtures of quaternary ammonium salts and metal chlorides, type II consists of a quaternary ammonium salt and a metal chloride hydrate, type III consists of a quaternary ammonium salt and a hydrogen bond donor compound (usually an organic molecular component, such as an amide, carboxylic acid or polyol), and type IV consists of a metal chloride hydrate and an HBD [20]. After describing a group of natural deep eutectic solvents, type V was created, in which the components are non-ionic organic compounds that are hydrogen bond acceptors and donors [18]. It is assumed that it is also possible to obtain a DES by mixing selected Brønsted–Lowry acids and bases, suggesting that further types of DESs can be determined [17]. The classification of deep eutectic solvents is shown in Table 1.

### 2.3. Natural Deep Eutectic Solvents

In 2011, Choi reported and described a new type of eutectic solvents, i.e., natural deep eutectic solvents. Choi’s research involved elucidating the solubility of intracellular compounds that are insoluble in both the aqueous and lipid phases [21]. The observations were later confirmed by Dai et al. in 2013, where it was shown that there are mixtures in cells containing combinations of metabolites that play key roles in biological processes such as cryoprotection, drought resistance, germination and dehydration, and the discovered mixtures may be the third liquid phase in living organisms [22].

Natural deep eutectic solvents are a group of deep eutectic solvents that contain natural basic metabolites, including sugars, sugar alcohols, carboxylic acids, amino acids and amines. Similarly, to DESs, the melting point of NDESs is lower than the melting point of each component and the components are linked to each other by hydrogen bonds, with a minimum of one component acting as a hydrogen bond acceptor and another as a hydrogen bond donor. Hundreds of mixtures of NDESs consisting of natural products, such as organic acids, alcohols, sugars, cholines, urea and its derivatives and amino acids, were already obtained [20]. Crucially, these compounds are relatively cheap, especially compared with ionic liquid reactants. For example, the popular choline is a component of vitamin B and is currently produced in megatons per year as a dietary supplement for livestock, while urea, a popular choice of HBD, is commonly used in fertilisers. A summary of the NDES components that are most used in nanotechnology is shown in Figure 3.

Several bonds and interactions are formed between NDES components, including hydrogen bonds, ionic attractions between cationic/anionic groups and van der Waals interactions [17]. The presence of the bonds allows for decreasing the melting point of the eutectic mixture. However, while the phase behaviour of DESs can be easily represented by phase diagrams, it becomes problematic to describe the phenomena with thermodynamic models. Present models, which do not take into account all the actual interactions that occur between the molecules, do not allow for predicting which compounds are able to form NDESs [23,24]. Therefore, the processes for obtaining NDESs are based on experimental methods. If the potentials of DESs and NDESs are to be realised, research and attempts to theoretically describe the modes of interaction of their components are necessary [25].

## 3. Synthesis of Natural Deep Eutectic Solvents 

One of the greatest advantages of natural deep eutectic solvents is the wasteless process of obtaining them. The process of synthesis of NDESs involves combining all components to form a homogeneous liquid. As a result, there is no need for additional solvents and the solvent formation process does not occur via a reaction with the formation of entirely new components. The reagents can be in either a liquid or solid phase, plus there is no need for prior purification. The combining of the components can, most commonly, occur via mechanical rubbing or by heating the mixture. The laboratory methods used for NDESs include vacuum evaporation, grinding, freeze-drying and mixing in the presence of an external energy source. A comparison of the methods for selecting the heating sources of reactants for obtaining NDESs is shown in Figure 4.

### 3.1. Mechanical Methods

The mechanical method involves grinding the NDES components together. At the laboratory scale, mortars are used, and it is possible, at a larger scale, to use ball mills [26]. Through the presence of various mechanical forces, such as shear, fracture and vibration, an efficient bonding process of the NDES components can be achieved, particularly if all components are in a solid state of aggregation [27]. A mechanochemical synthesis for the preparation of choline chloride (ChCl) with urea (choline chloride urea = reline) was described by Crawford et al. Using a twin-screw extrusion method, it became possible to combine the components while simultaneously grinding the ingredients. In addition, the method was a continuous process, making the process yield four orders of magnitude higher than when using batch methods [28].

### 3.2. Vacuum Evaporation and Lyophilisation

Vacuum evaporation is an alternative method of NDES synthesis. It involves dissolving the NDES components in a solvent (usually water) at room temperature. Once a solution is obtained under reduced pressure, the water is evaporated. The mixture with the remaining water content is placed in a desiccator to achieve a constant product weight. The lyophilisation process also uses an addition of about 5% mass water, in which the ingredients are dissolved. The systems are then frozen and freeze-dried to form a clear liquid [17].

### 3.3. Thermal Methods

#### 3.3.1. Conventional Heating

Conventional heating is the most used source of energy supply to a system that allows for natural deep eutectic solvents to be obtained. Reasons for using conventional heating include its availability, the gradual heating of the system and the ability to heat variable volumes of materials [29]. The conventional heating process involves mechanically or magnetically mixing the solid or liquid NDES components and heating the mixture to 40–100 °C until the compounds are completely dissolved and the NDES is formed [19]. 

#### 3.3.2. Ultrasound

An effective way to supply energy to the system is to use ultrasound. The advantage of such a solution is that the time to obtain an NDES can be reduced from several hours to even tens of seconds. However, the use of ultrasound requires a semi-fluid environment in which sound waves can propagate. Therefore, this solution will work when at least one component is a liquid. An alternative solution may be to introduce an already prepared NDES obtained via conventional methods or after partial fragmentation of the components by grinding. A limitation may also be the occurrence of local overheating, which may cause partial disintegration of particularly temperature-sensitive raw materials. However, these limitations do not detract from the potential of the presented method [30,31].

#### 3.3.3. Microwave Radiation

In recent years, there have been significant developments in the use of microwave radiation as an energy source in the synthesis of several compounds, both organic and inorganic. Furthermore, in the synthesis of natural deep eutectic solvents, methods using the microwave radiation field as an efficient energy source can be found.

As with ultrasound, microwave-assisted syntheses are faster and more efficient compared with conventional heating. The challenge with the use of microwaves is the use of an additional means of mixing the reactants, which is particularly important when combining and contacting NDES components [1]. On the other hand, the advantage over the rest of the methods is the possibility of increasing the scale while maintaining the efficiency of the energy input to the system.

Balaji et al. compared the possibility of obtaining natural deep eutectic solvents based on malic acid–citric acid–water and xylitol acid–malic acid–water, in each case using a molar ratio of components of 1:1:10. The study compared three different energy sources: controlled heating and stirring, ultrasound-assisted synthesis and synthesis conducted in a microwave radiation field. The process with conventional heating was carried out for 2 h, heating the reaction system to 50 °C at a speed of 220 rpm. In contrast, by using an ultrasonic bath and conducting the process in a microwave radiation field, the processing time was reduced to 45 min. The temperature in the microwave reactor was 80 °C, operating at 850 W and 10 bar pressure. Taking into account the power of the equipment, the synthesis time and the volume of the solvents, the energy consumption per synthesis was determined. The energy consumption for the synthesis of eutectic solvents was 0.014 KWh/cm^3^ for heating and stirring, 0.106 KWh/cm^3^ for microwave assistance and 0.006 KWh/cm^3^ for ultrasonic assistance. In terms of energy use, ultrasound-assisted synthesis is the most environmentally friendly, using approximately 57% less energy compared with the heating and stirring method [5]. Importantly, the authors observed no differences in the properties of the NDES obtained under various energy sources, confirming the possibility of using different methods in the synthesis of natural deep eutectic solvents. 

## 4. Properties and Characterisation of Natural Deep Eutectic Solvents

A variety of organic compounds can be selected as NDES components. The choice of ingredients will have a key impact not only on the physical properties of NDESs but also on their chemical properties and, consequently, will affect their applicability [32,33]. This review describes major properties that are particularly relevant to the properties of nanomaterials. In addition to the basic physicochemical properties, i.e., density, viscosity and melting point, in the preparation of nanomaterials and their modification, i.e., refractive index, reducing activity and stabilising properties are also important. In the following subsections, the main properties of natural eutectic solvents are described, highlighting their importance in nanotechnology. An extensive description of the properties of DESs can be found in the literature [16,17,25].

### 4.1. Viscosity and Density

The viscosity of NDESs is significantly higher than that of most typical molecular solvents, and the temperature-dependent change in viscosity varies in an Arrhenius manner with large activation energies for viscous flow. Due to the structure, according to hole theory, NDESs composed of ionic compounds contain voids subject to steady flow after melting. The hole theory and ionicity have been extensively described in the literature [16]. Holes have a random size and position, with the assumption that their average size is similar to that of the matching ion. The smaller the ion is, the easier it is to move to a vacant spot, provided that the cavity is of the correct size in the surrounding volume. As the temperature decreases, the average size of the holes will decrease, which contributes to limiting their mobility, which is why the viscosity of NDESs tends to take on such high values [34]. It is assumed that the limiting factor in fluid viscosity is not the thermodynamics of hole formation, but the probability of locating voids. This approach facilitates the modelling of fluid viscosity, as well as the prediction of NDES conductivity. The commonly used equations include the Stokes–Einstein equation, the Arrhenius equation and the Vogel–Fulcher–Tammann (VFT) equation [17]. Using selected models, the temperature dependence of viscosity can be determined. Increasing attention is being paid to these issues in the literature, enabling a higher degree of description and prediction of the behaviour of a substance [35]. Based on the conventional Arrhenius temperature dependence, the empirical Vogel–Fulcher–Tamman (VFT) equation was derived. The VFT equation is used to describe the viscosity of a fluid as a function of temperature and, in particular, its strong temperature dependence when approaching the glass transition. The Stokes–Einstein equation describes the relationship between the intrinsic diffusivity of a substance and its hydrodynamic radius in a viscous medium, taking into account the thermal energy required from the particles to overcome the viscous force of the medium through which they move [36].

The ability to model the behaviour of NDESs was confirmed by Aroso et al. [37]. By analysing the flow curves for mixtures of choline chloride or betaine with sugars in the shear rate range from 0.1 to 100 s^−1^ at 283–373 K, the authors confirmed the Newtonian behaviour of the liquids and obtained a high degree of fit of the experimental data to the Arrhenius theoretical model. In practice, most NDESs exhibit a high viscosity, which is characterised by the “syrupy nature” of their flow. High viscosities imply limitations in their use in practice. For example, the viscosity of ethaline is 52 cP, compared with 1 cP for water (at 20 °C). It is possible to change the viscosity of materials either by increasing the temperature of the reaction system or by modifying the composition of the NDES. However, both methods are not always feasible or cost-effective. A broad description of the physicochemical correlations with temperature, including viscosity, conductivity and density, as well as a description of hole theory are described in the literature [38,39,40,41]. The models aim to explain ion mobility in high-temperature molten salts and deep eutectic solvents and are based on the observation that the volume of these liquids increases after melting. Crucially, however, this approach is suitable for predicting the behaviour of previously well-understood substances of known composition in which the density, viscosity and other basic properties have been experimentally determined [42].

Density provides information about intermolecular interactions in NDESs. As with viscosity, mixtures of NDESs typically exhibit a density higher than that of water (e.g., ethaline has a density of 1.14 g cm^−3^ and glycerine has a density of 1.19 g cm^−3^ at 20 °C). Reducing the density is possible by modifying the composition of NDESs, but this translates into changing their other physicochemical properties as well. Basaiahgari et al. measured the density of ethylene, diethylene, triethylene glycol and glycerol as HBDs and benzylammonium chloride salt as an HBA. Their results showed that ethylene glycol (EG)-based DESs had a lower density compared with glycerol-based DESs. The increase in density after replacing ethylene glycol with diethylene glycol, triethylene glycol and glycerol implies that increasing the number of -OH functional groups on the HBD results in the formation of more H-bonds, which presumably reduces the available volume [43]. Modification of the density or viscosity of NDESs can occur not only by replacing one component with another but also by changing the molar ratio of HBAs to HBDs. Shafie et al. presented the densities of different molar ratios of ChCl to citric-acid-based DESs. They found that as the ratio of ChCl to citric acid increases, the density decreases [44].

### 4.2. pH of Natural Deep Eutectic Solvents

NDESs that predominantly belong to DES type III consist of a mixture of hydrogen bond acceptors and donors. This translates into the variable pH values that NDESs can adopt. The composition of NDESs, namely, the proportion and types of components, will be key to the changes in pH. The acidity of the mixture is relevant, particularly in the preparation of nanoparticles, including the kinetics of the chemical reactions taking place [17]. Metal ions form complexes with the components of NDESs. In addition to the influence of basic H^+^ and OH^-^ ions, the complex forms that form during the processes will be of dominant importance for the processes in NDES systems. The study of metal ion speciation in DESs is still a subject of research. The complexity of the problem, which is due to the presence of various Lewis anions with different alkalinity, as well as the varying compositions of DESs, make it challenging to understand in detail. However, understanding speciation in DESs is extremely important when trying to determine the mechanisms of nucleation and growth of inorganic nanoparticles [16].

### 4.3. Refractive Index

The refractive index (R_I_), as a dimensionless property of a material, is particularly important in the context of obtaining suspensions of metallic and non-metallic nanoparticles. The refractive index R_I_ determines how much the speed of light changes when passing through a medium relative to the speed of light in a vacuum. The R_I_, which is specific to each solvent, must be taken into account when determining the size of nanoparticles using dynamic light scattering (DLS). It is therefore a useful tool that complements the measurements of physical properties. Its values vary depending on the type of components used, as well as their molar ratios in the NDES [17]. The refractive index range has values of approx. 1.33 to approx. 1.59, depending on the composition and water content [45,46]. For example, the RI for ethaline (choline chloride with ethylene glycol 1:2) and glyceline (choline chloride with glycerol 1:2) at 298.15 K are 1.468 and 1.487, respectively [47].

### 4.4. Reducing Properties

In contrast to organic solvents or water, NDESs, due to their composition, often exhibit strong redox properties. This property can be particularly useful in processes used to obtain nanoparticles. The redox potential of such materials can be determined using cyclic voltammetry (CV) among other techniques [20,48]. CV is a widely used electrochemical technique in which cyclic voltamperograms with two characteristic peaks are presented [49]. The diagrams represent a reversible redox process (Figure 5). The anodic current peak corresponds to the anodic oxidation of the analyte and the cathodic current peak is associated with the reduction process of the oxidation product observed in the return cycle after a change in electrode polarity [50]. The estimation of the standard electrochemical potentials of oxidation and reduction comprises one of the most widespread applications of CV, making the technique popular in NDES analysis [51].

Alnashef et al. investigated the electrochemical behaviour of iron (III) acetylacetonate in six different deep eutectic solvents formed via hydrogen bonding between ammonium and phosphonium salts with glycerol, ethylene glycol and triethylene glycol. Cyclic voltammetry was used by the authors to determine the kinetic and mass transport properties of the electrolytes, including determining the diffusion coefficient of the iron salts and the electron transfer rate constants, which allow the redox properties of the components to be described for both oxidation and reduction processes [53]. Based on cyclic voltammetry curves, Xu et al. indicated that the diffusion coefficient of iron ions in the ethaline DES system was higher than that in the reline DES system. These results may be useful in evaluating the reducing potential of natural deep eutectic solvents in the reduction of metal ions to nanoparticles [54]. Based on cyclic voltammetry and electrochemical processes, Baby et al. investigated the effect of different deep eutectic solvents on the synthesis of MgFe_2_O_4_ nanoparticles, which were to be used for the electrochemical determination of nitrofurantoin and 4-nitrophenol in a subsequent step. Depending on the NDES composition used, different reduction peaks were obtained on the graphs, which corresponded to different redox properties [55].

### 4.5. Complexing and Stabilising Properties 

Surface tension determines the tendency of a material towards a minimum surface area. Its effects are most prevalent in liquids due to intermolecular interactions between molecules in a liquid, and this translates into stability and reduced agglomeration. Surface tension measurements can show which compounds act as surfactants and reduce cohesion forces, causing NDESs to impart stabilising properties to the forming nanoparticles [17]. Gajardo-Parra et al. measured the surface tension of ChCl-based DESs with levulinic acid, phenol and ethylene glycol. The authors found that the surface tension of ethaline was 45.66 mN/m at 25 °C and lower than that of pure ethylene glycol (48.90 mN/m), confirming that ChCl acts as a surfactant and reduces the cohesion forces on the surface of ethaline [56].

An alternative approach is to select the composition of the NDESs in such a way that the content of compounds exhibiting stabilising properties is maximised. However, it remains to be seen that the stabilisation of nanoparticles can proceed along several pathways, including via a steric effect, using large highly extended stabilisers and charge stabilisation via the deposition of compounds with selected surface groups, e.g., strongly hydrophilic/hydrophobic or positively/negatively charged [57]. In the preparation of inorganic nanoparticles, compounds used as stabilisers that can also be successfully used as components of NDESs include carboxylic acids (citric acid, maleic acid, ascorbic acid) and sugars (glucose, sucrose, galactose, mannose).

### 4.6. Thermal Properties

Knowing the eutectic point allows NDESs to be used with a composition that has the lowest melting point of the product. This results in the ability to operate at lower temperatures and a wider temperature range, affecting the viscosity of the NDES and their other physicochemical parameters. For example, the eutectic point of reline occurs at a choline chloride–urea molar ratio of 1:2. Unfortunately, details of the eutectic composition of individual DESs are determined experimentally, and thus, the number of papers describing this issue is limited [17].

Thermogravimetric analysis (TGA) enables the detection of deviations and changes in behaviour due to events such as phase changes, the addition of additional reactants and the analysis of product stability over time. TGA is used in DES to understand fundamental information about thermal behaviour in both crystalline and glassy transition states. Knowing the behaviour of NDES systems as a function of temperature and time provides insight into the many different physicochemical properties that occur during absorption, desorption, sublimation, evaporation, decomposition, oxidation and reduction, dissolution, etc. [20,48]. An example of the use of thermogravimetric analysis is obtaining thermal decomposition profiles of synthesised solvents, which provide evidence of the occurrence of interactions between precursors and of changes occurring in processes carried out at temperatures higher than 150 °C, which occurs, for example, in the preparation of metal oxides.

A complementary thermal method is differential scanning calorimetry (DSC), which is a thermoanalytical technique that measures the amount of heat required to produce an observed temperature change in a sample. This analysis is used to determine melting points, enthalpies of formation and melting, heat capacity and thermal stability. It is possible to compare changes in these parameters of NDES components as well as the resulting mixtures. The ability to capture phase transitions makes DSC particularly useful for detecting anomalies in DESs and determining the post-process behaviour of NDESs by determining the possibility of recycling these compounds back into the process [17].

## 5. Natural Deep Eutectic Solvents Applications in Nanotechnology

Since 2003, i.e., since Abbott’s presentation of the new material group of deep eutectic solvents, DESs have become a convenient and environmentally friendly alternative to aqueous conventional solvents or ionic liquids [6]. Through the additional properties that NDESs possess, their applications are constantly expanding. Currently, the main research efforts are concentrated in areas such as biomedicine, metal treatment and extraction of a series of compounds, metallurgy, electrodeposition, separations, gas capture and biocatalysis [15,20,58,59,60]. Nanotechnology is a new field in which NDESs may find an application (Figure 6).

Five years after the announcement that DESs exist, Sun et al. presented the first study in which they described a method for obtaining NDES-assisted gold nanoparticles based on choline chloride and urea [7]. DESs have so far found applications as reaction media for the synthesis of nanomaterials, for the electrodeposition of nanomaterials, as dispersing agents, as nanoparticles or modifiers affecting the morphology or chemical composition of nanoparticles, and as substances affecting the nucleation and growth of nanostructures. Among the inorganic materials obtained in the presence of NDESs, metal oxides and metals, especially gold nanoparticles, are most commonly mentioned. Descriptions of methods for obtaining their metal nanoparticles can be also found, but these processes are more problematic to carry out.

### 5.1. Reaction Media for the Synthesis of Nanomaterials 

Natural deep eutectic solvents, as substances of organic origin that remain as liquids at room temperature, are excellent materials in which reactions can be carried out. Their main advantage is their liquid form without the use of water or simple organic solvents [20,44,56,58]. This property is typically exploited in processes to obtain suspensions of metal nanoparticles. Additionally, depending on the choice of NDES, we can base the solution on specific parameters, i.e., viscosity, pH, refractive index, etc., which, depending on the future applications of the material, can be crucial. The composition of NDESs and their form, especially at the beginning of their applications in nanotechnology, was used as a reaction medium in electrochemical processes to obtain a range of nanomaterials [60,61].

In the descriptions of NDESs, one of the main properties that is given is that NDESs are classified as a type of green solvent. NDESs are often used as solvents and extraction solutions, but in some cases, they can also be used as reactants to produce the intended nanoparticles [58]. Among the applications of NDESs is their use as reaction media for the preparation of nanoparticles, e.g., calcium phosphate, hydroxyapatite and fluorapatite. Particularly in the development of biomedical materials, conducting reactions in NDES perfectly suits the purpose. Based on a choline chloride/urea mixture, the preparation of calcium phosphate or hydroxyapatite, among others, enabled the control of particle size, as well as produce elementally and structurally highly pure crystalline products with good biocompatibility and mineralisation ability. An important benefit of using NDESs is that they can be recovered and reused in the process. After synthesis, the DESs were recovered and reused for the synthesis of hydroxyapatite nanoparticles [62]. In another study, Anicai et al. used ChCl/EG and ChCl/urea as the reaction medium to obtain TiO_2_ nanoparticles. By using such a reaction system, it was possible to obtain a TiO_2_ nanopowder through electrochemical synthesis.

Among others, NDESs have found applications in the recently developed ionothermal synthesis in which the DES or ionic liquid is the solvent and template, i.e., the structure-directing agents. The possibility of using these compounds is due to their low vapour pressure, enabling the ionothermal process to be carried out in a low-pressure environment, i.e., at ambient pressure. Eliminating the need for autoclaves increases safety by simplifying the synthesis process. Moreover, as a result of the possibility to tune their ionic character, NDESs can act in a dual role, both as a reaction environment and as a surface modifier of nanomaterials by being the source of the required functional groups. This enables an ‘in situ’ modified material to be obtained with the required surface and desired functional properties. The number of methods based on the ionothermic process using NDESs to produce nanoparticles is increasing every year. In their work, Xiong et al. proposed an ionothermal method using ChCl/urea as the reaction media to prepare Fe_2_O_3_ nanoparticles, which confirmed the effectiveness of using NDESs as a reaction medium [63]. 

### 5.2. Reducers and Stabilisers for Nanomaterials

Depending on the compounds included in their composition, the use of NDESs can allow the morphology of the nanomaterials produced to be directed. Influencing the size and shape of the nanoparticles allows for materials with the required properties to be obtained and makes it possible to increase their applicability. Changes can occur both during the production reaction and afterwards for dispersion in colloidal applications. These can occur by contacting the forming nanoparticles with specific functional groups of the NDES compounds. NDES parameters, such as viscosity and tension, prevent aggregation and agglomeration of the nanoparticles such that their stability in the NDES is maintained [64].

Oh et al. used DESs with ChCl and malonic acid both as a reaction medium and a structure-directing agent for the synthesis of gold nanoparticles. The composition of the NDESs allowed for nanoparticles with a highly defined diameter to be obtained. The synthesis was not supported by any surfactants or polymers, suggesting that DES plays an essential role as a structure-directing reagent and particle stabiliser [65].

### 5.3. Biocompatibility of NDESs—Applications in Medicine

As the name suggests, NDESs are based on compounds of natural origin, mainly primary plant metabolites, and can be used in fields such as medicine, cosmetology or food production [59,66]. Even when prepared using industrial methods, they show high biocompatibility and are biodegradable and non-toxic. As a result, they can be successful green replacements for compounds that are currently used, e.g., stabilisers, such as PVP and PVA; used reductants, such as sodium borohydride or hydrazine; or solutions, such as certain organic solvents.

In 2015, the concept of therapeutic deep eutectic solvents (THEDES) was even introduced, which are compounds that can be successfully used as auxiliary materials in the delivery of drugs and active substances [67]. For example, Makkliang et al. used choline chloride–propylene glycol (1:2) at 56.1 °C for a cellulolytic enzyme reaction in which daidzein and genistein were extracted and completely converted to their aglycones. Biocompatible DESs enhance the activity of the cellulolytic enzyme, producing compounds with higher bioavailability [68]. Zhang prepared magnetic molecularly imprinted nanoparticles with a deep eutectic solvent for medical applications to separate transferrin in human serum. The authors confirmed that the use of DES provides an efficient and biocompatible method for protein isolation and purification [69].

### 5.4. Examples of Preparation of Nanoparticles Using NDESs

Since the first description of deep eutectic natural solvents, the number of applications in nanotechnological fields has increased year by year. The materials can be used to prepare well-defined nanoparticles of controlled shape and size, including obtaining their various forms as films and coatings, colloidal suspensions or powder materials, among others.

On the one hand, one may observe increased viscosity compared with water or standard organic solvents, which is a disadvantage of NDESs. On the other hand, it promotes the possibility of dispersion formation of nanoparticles, not allowing rapid growth to a macrocrystalline form, while maintaining their stability. The mechanisms of nucleation and growth of nanoparticles are strongly dependent on the composition of the DES. The choice of components influences the production of a material with the desired reducing potential, and the presence of selected groups and compounds in the NDES system enables the preferential growth of crystals [11].

The literature describes processes for the preparation and modification of inorganic nanoparticles, in which NDESs are the raw materials, reaction medium, reducing agent, stabiliser or surface modifier. Methods for the synthesis of metal nanoparticles, metal oxides, sulphides and salts that can successfully compete with their counterparts obtained by conventional methods are described.

#### 5.4.1. Examples of Obtaining Metal Nanoparticles

Methods for the preparation of gold, silver, copper, nickel and platinum nanoparticles are described in the literature. In their preparation processes, NDESs perform the role of, among others, the reaction medium, stabilising agent, reductant and surface modifier of nanoparticles. Table 2 shows the obtained nanoparticles of selected metals in natural deep eutectic solvents along with the roles of NDESs.

Gold nanoparticles were the earliest nanoparticles to be obtained using NDESs. In 2008, Liao et al. reported on the simple synthesis of Au NPs using ChCl/urea DESs as a stabilising agent. Importantly, the addition of water made it possible to obtain nanoparticles with variable shapes. The authors obtained star-shaped Au nanoparticles at a water content of 5000 ppm, Au nanorods at a water content of 10,000 ppm and snowflake-shaped particles in an anhydrous medium [7]. In contrast, Chirea et al. compared two different DESs, i.e., choline chloride–ethylene glycol (1:2) and choline chloride–urea (1:2), as stabilising agents for gold nanotubes from the direct reduction of HAuCl_4_ using NaBH_4_ [70]. Crescenzo et al. used NDESs based on betaine N,N,N-trimethylglycine and oxalic acid, both as stabilisers and reductants in the synthesis of Au NPs. It was only necessary to heat the system from 30 to 80 °C. In contrast, using other carboxylic acids (glycolic and phenylacetic acids), an additional reductant was necessary, but no elevated temperatures were required. This demonstrates that it is possible to control both the composition and the process conditions to obtain the desired nanoparticle structures [71]. The significant effect of temperature was also confirmed by Oh et al., who presented a description of how gold nanoparticles were prepared using choline chloride and malonic acid, which acted as a reaction medium, a stabilising agent and a growth-directing agent for the structure. Spherical Au NPs with a size of nearly 100 nm were synthesised at 70 °C, while lattice-like nanostructures were observed when the synthesis temperature was 90 °C [65].

Silver nanoparticles were initially difficult to obtain using the widespread presence of chlorine in NDESs. At present, however, they are successfully synthesised using NDESs. An example is the preparation of silver nanoparticles with a narrow size distribution of about 4.5 nm, which was prepared even using reline based on the laser ablation method of a metallic silver wafer [72]. Adhikari et al. developed an unusual synthesis of Ag NPs using AgCl as a silver precursor. Nanoporous Ag films on copper alloy substrates were obtained using a galvanic exchange reaction from a ChCl–EG solvent (1:2) containing AgCl, along with the morphology of the nanoporous films [73]. Adhikari et al. in another study proved the feasibility of obtaining Ag and Au metal nanoparticles at high metal concentrations (400 and 1000 mM, respectively) [3]. 

**Table 2 materials-16-00627-t002:** Metal nanoparticles obtained using NDESs with a description of the method of obtaining NDESs and the functions it performs in the system.

Material	Natural Deep Eutectic Solvents	Process Parameters of NDES Synthesis	Size of Nanoparticles	NDES Function	Ref.
Au	Choline chloride–urea (1:2)	Conventional heating in 100 °C	300 nm	Reaction medium, stabiliser	[7]
Au	Choline chloride–urea (1:2)	Conventional heating in 80 °C	5 nm	Growth media	[74]
Au	Choline chloride–urea (1:2) Choline chloride–ethylene glycol (1:2)	Conventional heating in 80 °C	17.74–23.54 nm	Reaction medium, stabiliser	[75]
Au	Choline chloride–gallic acid–glycerol (1:0.25:0.25)	Conventional heating in 100 °C	30–100 nm	Reaction medium, stabiliser, reducer	[76]
Au	trimethylglycine–glycolic acid Trimethylglycine–phenylacetic acid, trimethylglycine–glycerol	Conventional heating in 90 °C	4–7 nm	Reaction medium, stabiliser	[71]
Ag	Choline chloride–ethylene glycol (1:2)	Conventional heating in 50 °C	Ag coatings 35 nm thick	Material used to disperse the silver layer on the nickel substrate	[72]
Au	Choline chloride–urea (1:2)	Conventional heating in 100 °C	35.1–46.7 nm	Reaction medium	[77]
Ag	Choline nitrate–glycerol (1:2) Choline chloride–glycerol (1:2) Choline acetate–glycerol (1:2)	Conventional heating in 60–70 °C	7.6–11.5 nm	Reaction medium	[73]
Ag	Silver triflate–acetamide (1:4)	Conventional heating in 70 °C	9–16 nm	Reaction media, source of silver nanoparticles	[78]
Pt	Choline chloride–urea (1:2)	Conventional heating in 80 °C	Approx. 200 nm	Reaction medium, growth media	[79]

#### 5.4.2. Examples of Obtaining Metal Oxide Nanoparticles

The conventional preparation of inorganic materials, including metal oxide nanoparticles, is often carried out in water or organic solvents, in many cases using a necessary thermal step for crystallisation and nanostructure formation, such as hydrothermal, ionothermal or calcination processes [80]. It is possible to obtain nanoparticles of ZnO [81], SiO_2_ [82], Fe_3_O_4_ [83], Mn_3_O_4_ [84], SnO_2_ [85] and many others. Table 3 summarises the preparation of selected metal oxide nanoparticles in the presence of NDESs.

Among the methods for obtaining metal oxide nanoparticles is the antisolvent approach. After dissolving macrometric oxides in NDESs, a new solvent is added to precipitate new nanometric metal oxide forms. Dong et al. developed a method in which, using the choline chloride/urea system, they dissolved ZnO by adding water or ethanol and subsequently precipitated ZnO nanostructures of various shapes [86]. A separate method was presented by Chen et al. who, using choline chloride and succinic acid (1:1), applied anodic dissolution to titanium and obtained titanium oxide nanoparticles [87]. After the application of a choline dihydrogencitrate salt–oxalic acid, titanium oxide nanoparticles were also obtained, but with a different structure [88].

An alternative approach is a method in which nanoparticle precursors are precipitated in an NDES environment and calcined. Sçldner et al. developed a method for the solid-state preparation of phase-pure magnesium ferrite nanoparticles at 500 °C. The effect of five different DESs on the effectual formation, structure and composition of magnesium ferrite nanoparticles was verified [89]. The process involves the formation of magnesium ferrite structures in selected DESs, followed by their calcination, which allows for the formation of regular structures. The process of obtaining ferrite nanoparticles was also described by Das et al., who obtained MgFe_2_O_4_, ZnFe_2_O_4_, CoFe_2_O_4_ and NiFe_2_O_4_ nanoparticles. In the first step, the iron oxides were dissolved in ChCl/maleic acid with the oxides of the other metal. After 2 h of stirring, calcination processes were carried out (at 400, 500 or 600 °C) to obtain final nanoparticles with sizes of 120–480 nm, depending on the process conditions [11].

A large proportion of the methods in which metal oxide nanoparticles are obtained using NDESs are based on ionothermal processes, including MnO_x_ [90], FeCo LDH [29], Fe_2_O_3_ [63] and Fe_3_O_4_ [91]. Hammond et al. observed that in the ionothermal process, reline acts as a catalyst to combine reactive components in the presence of water, i.e., Ce(NO_3_)_3_-6H_2_O or CeCl_3_, enabling the preparation of CeO_2_ nanoparticles [92].

Using the NDES system with the addition of a reductant, it is possible to obtain nanoparticles, including metal oxide nanoparticles. Using choline chloride ethylene glycol as the solvent and hydrazine hydrate as the reducing agent, Balaji et al. obtained nanoparticles of ZrO_x_, MnO_x_ and CuO, among others. Importantly, the use of a reducing agent made it possible to limit the use of high temperatures, as metal oxide nanoparticles were obtained at temperatures as low as 50 °C. However, it was necessary to extend the reaction time to 2 days [93]. 

**Table 3 materials-16-00627-t003:** Metal oxide nanoparticles obtained using NDESs with a description of the method of obtaining the NDESs and the functions they perform in the system.

Material	Natural Deep Eutectic Solvents	Process Parameters of NDES Synthesis	Size of Nanoparticles	NDES Function	Ref.
Ag_2_O	Malonic acid–glucose (1:1) Malonic acid–fructose (1:1)	Evaporation in a vacuum at 50 °C	95.42–185.99 nm	Reaction medium, stabiliser	[94]
CeO_2_	Choline chloride–urea (1:2) Choline chloride–glycol ethylene (1:2)	Conventional heating at 80 °C	10–30 nm	Ionothermal reaction medium	[92]
Cu_2_O	Choline chloride-ethylene glycol (1:2)	Conventional heating at 100 °C	116.2–148.2 nm	Reaction medium, stabiliser	[95]
Cu_2_O on TiO_2_	Choline chloride and urea (1:2)	Conventional heating at 60 °C	1.2–2.0 nm	Reaction medium	[96]
Fe_2_O_3_	Choline chloride–urea (1:2)	Conventional heating at 80 °C	180 nm	Ionothermal reaction medium	[63]
Fe_3_O_4_	Choline chloride–urea (1:2)	Conventional heating at 80 °C	11 nm	Reaction medium, stabiliser	[83]
Mg, Co, Ni ferrite	Choline chloride–maleic acid (1:1) Choline chloride–oxalic acid (1:1) n,n’-Dimethylurea–citric acid (7:2)	Heated up to 80 °C in an aluminium heating block	120–480 nm	Reaction medium, stabiliser	[89]
Mn_x_O_y_	Choline chloride-glucose (1:2) Choline chloride–ethylene glycol–glucose (1:1:1)	Conventional heating at 58–60 °C	1300 nm	Reaction medium, stabiliser	[97]
Mn_3_O_4_	Choline chloride–ethylene glycol (1:2)	Conventional heating at 90 °C	18 nm	Reaction medium, stabiliser	[84]
NiO	Choline chloride–urea (1:2)	Conventional, 80 °C	100 nm	Reaction medium	[98]
SnO_2_	Choline chloride–urea (1:2)	Conventional, 80 °C	4–10 nm	Reaction medium	[85]
TiO_2_	Choline chloride–succinic acid (1:1)	Conventional heating at 80–100 °C	40–65 nm	Reaction medium, stabiliser	[87]
ZnO	Choline chloride–urea (1:2)	Mixing in a glovebox with an argon atmosphere	60–650 nm	Reaction medium, stabiliser, solvent of ZnO	[86]
ZnO	Choline chloride–ethylene glycol (1:2)	Conventional heating at 80 °C	30 nm	Ionothermal reaction medium	[81]
ZnO	Betaine and phenol (1:2)	Conventional heating at 30–40 °C, 30 min	100 nm	Reaction medium	[13]
ZrO_x_ MnO_x_ CuO	Choline chloride–urea (1:2)	Conventional heating at 60 °C, 1 h	19.16 nm 49.46 nm 44.22 nm	Reaction medium	[93]

#### 5.4.3. Examples of Obtaining Salt Nanoparticles and Other Inorganic Nanoparticles

In addition to obtaining metallic and oxide nanoparticles, it is possible to obtain more complex structures, e.g., sulphides, oxygen salts, double hydroxides, nanocomposites (such as ZnCo_2_O_4_) [99], nickel/nickel nitride nanocomposites [100] and bismuth vanadate microtubes (BiVO_4_) [101] (Table 4). An example of the preparation of sulphides was presented by Mohammadpour et al., who successfully obtained MoS_2_ nanosheets. For this purpose, they used different NDES mixtures with sugars (glucose, fructose, sucrose), choline chloride and water [102]. Moreover, based on the ionothermal method, sulphide nanoparticles can be obtained, as confirmed by Chen et al. [87]. The authors prepared thin films based on lead sulphide (PbS) nanotubes using a ChCl/urea DES. A study by Zhang et al. presented a versatile method for obtaining a range of sulphides (Sb_2_S_3_, Bi_2_S_3_, PbS, CuS, Ag_2_S, ZnS and CdS) using a mixture of choline chloride and thioacetamide acting as a solvent, reactant and stabiliser. In a two-step process, a metal–DES complex was formed by adding suitable metal salts to the DES solution to decompose the metal–DES complex into metal sulphides upon heating [103].

Phosphates are examples of salt nanoparticles that can be obtained in an NDES environment. Using NDESs further increases their biocompatibility, which is particularly beneficial. Liu et al. developed a method to obtain zinc phosphate nanoparticles in the presence of choline chloride with imidazolidone (1:2) [104]. Based on the ionothermal method, Liu et al. obtained zirconium phosphate. The DES consisted of tetramethylammonium chloride/urea or oxalic acid dehydrate [105]. Hydroxyapatite structures with enhanced biocompatibility were obtained using choline chloride–urea as a solvent [106]. The eutectic choline chloride–urea mixtures used as solvents controlled the particle size, as well as leading to elemental and structurally highly pure crystalline products with good biocompatibility while maintaining solvent recoverability.

**Table 4 materials-16-00627-t004:** Inorganic nanoparticles obtained using NDESs with a description of the method of obtaining NDESs and the functions they perform in the system.

Material	Natural Deep Eutectic Solvents	Process Parameters of Synthesis of NDES	Size of Nanoparticles	NDES Function	Ref.
Amorphous Ca_3_(PO_4_)_2_	Choline chloride–urea (1:2) Choline chloride–ethylene glycol (1:2) Choline chloride–glycerol (1:2)	Conventional heating at 100 °C	24–39 nm	Reaction medium, stabiliser	[62]
MgFe_2_O_4_	Choline chloride–malonic acid (1:1) Choline chloride–oxalic acid (1:1) Choline chloride–urea (1:2) Choline chloride–ethylene glycol (1:2) Choline chloride–fructose (2:1)	Conventional heating at 25–80 °C	<100 nm	Reaction medium, stabiliser	[55]
(Mg, Sr, Zn)_3_(PO_4_)_2_	Choline chloride–fructose-H_2_O (5:2:5) Choline chloride–glucose-H_2_O (5:2:5) Choline chloride–sucrose-H_2_O (4:1:4)	Heated at 80 °C with continuous sonication	<500 nm	Reaction medium	[107]
CuCo_2_O_4_	Choline chloride–urea 1:2	Conventional heating at 60 °C	10–45 nm	Reaction medium	[108]
FeMnO_3_	Thymol–menthol (1:1)	Conventional heating at T_pok_, 30 min	<100 nm	Reaction medium	[109]
HgS	Choline chloride–urea (1:2)	Conventional heating at 60 °C, 1 h	23.51 nm	Reaction medium	[93]
CuCl	Choline chloride–urea (1:2)	Conventional heating at 74 °C	50 nm	Reaction medium	[110]
(Ni(NH_3_)_6_Cl_2_, NiCl_2_, α-Ni(OH)_2_ and NiO	Choline chloride–urea (1:2)	Conventional heating at 80 °C	Approx. 100 nm	Ionothermal reaction medium, nucleation and growth control	[111]

## 6. Current Limitations of Application of NDESs in Nanotechnology

This article presents applications in which natural deep eutectic solvents performed well. Over the 20 years in which these materials have been described, their importance and applicability has increased significantly [64]. However, there are still many issues that researchers will need to solve in the near future. Despite the great potential that natural deep eutectic solvents have, they have some limitations that, without a solution, will not allow for the widespread use of NDESs. The main disadvantages of NDES include the difficulty in significantly scaling up their preparation, the viscosity and density of the materials and the prediction of their properties using theoretical methods [112].

During the preparation of NDESs, some difficulties occur when transferring from the laboratory scale to the technical and industrial scale. Due to the need to mix solid compounds, the mass and energy transport correlations do not change directly proportionally. Due to the bonding processes that take place between hydrogen bond donors and acceptors, it is necessary to achieve a high degree of mixing and efficient energy exchange in the system. The solution may be to make the process leaner or to use alternative sources of heating and mixing of the reaction systems. Adhikari et al. presented a flow-through method for obtaining gold and silver nanoparticles using NDESs from dimethylammonium nitrate and polyol. The authors simultaneously presented solutions to two challenges related to the processes of obtaining NDESs, as well as obtaining metal nanoparticles with high concentrations (in the study, they obtained Au and Ag suspensions with metal concentrations of 400 and 1000 mM, respectively, and then converted this approach into an automated millifluidic continuous flow reaction format [3]).

Despite relying on basic widely available ingredients to obtain NDESs, the cost of their production is relatively high compared with water or alternative organic solvents. It is therefore necessary to verify the recyclability of NDES after nanoparticle preparation processes to reuse them. Several studies reported successfully recycling DESs after their use as dispersion media and in the production of nanoparticles and nanocomposites [113,114]. Yan et al. used NDESs consisting of choline chloride and oxalic acid to pretreat maize cob. Using oxalic acid regeneration and lignin removal, even after ten recycling cycles, the efficiency of the DES pretreatment did not significantly reduce glucan digestibility and glucose recovery (66.23% and 64.43%, respectively) compared with the original DES pretreatment (72.83% and 68.83%, respectively) [115].

Another barrier to the widespread use of NDESs is the ability to obtain unlimited combinations of solvent compositions. The possibility of combining compounds with such different properties has many advantages but determining at what molar ratio it is possible to obtain a deep eutectic solvent is only possible via experimental means. It is therefore necessary to develop models that can predict the behaviour of mixtures and verify whether such systems can be combined. The development of predictive models is the key to realising the full potential of NDESs. According to Hansen et al. this can be achieved through parallel efforts, including the following [17]:(a)Experiments that explore potential interrelationships between the properties of NDES components;(b)Carefully collecting, cataloguing and publishing all possible/practical physicochemical properties, especially for commonly studied compounds and constituents of DES (using reproducible synthesis protocols, carefully controlled storage of samples, detailed treatment, and pretreatment methods, etc.);(c)Processing such aggregated data and applying advanced computational techniques to find new correlations or empirical fits;(d)Undertaking much more in-depth studies on the coupling of liquid-phase dynamics to physical properties;(e)Taking a more detailed approach to understanding the nature and behaviour of esoteric hydrogen bond types/networks, particularly regarding the unusual behaviour of DES.

## 7. Conclusions

In this review, the potential role of natural deep eutectic solvents in nanotechnology is discussed, particularly in preparation processes for inorganic nanoparticles. Due to the requirements for more environmentally friendly technologies, the development of alternative methods for nanomaterials synthesis while reducing energy consumption and replacing conventional reactants with less toxic ones has become a priority goal for the scientific and production communities. Natural deep eutectic solvents can become a valuable alternative, as they can be easily prepared, can have reducing and stabilising properties, are capable of modifying the properties of the nanoparticles obtained and provide a reaction medium. There has been a steady increase in literature publications describing the use of deep eutectic solvents, including in nanotechnology. This article presents the physicochemical properties of NDESs, which are used to obtain functional nanomaterials, including metals, metal oxides and salts, and describes examples of NDES applications and the functions they perform in obtaining nanoparticles. An overview is also included of the issues and challenges that are still unresolved by the scientific world. 

## Figures and Tables

**Figure 1 materials-16-00627-f001:**
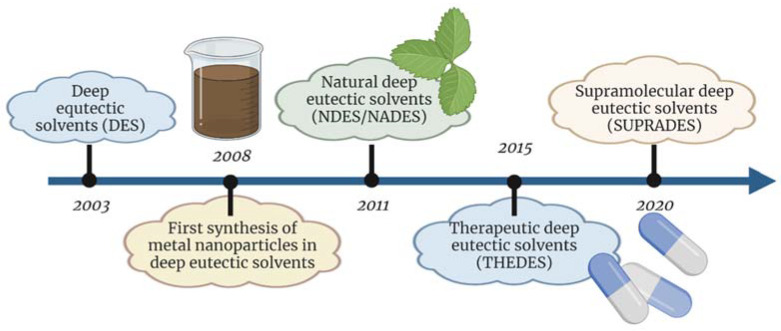
Historical milestones of deep eutectic solvents (created with BioRender.com).

**Figure 2 materials-16-00627-f002:**
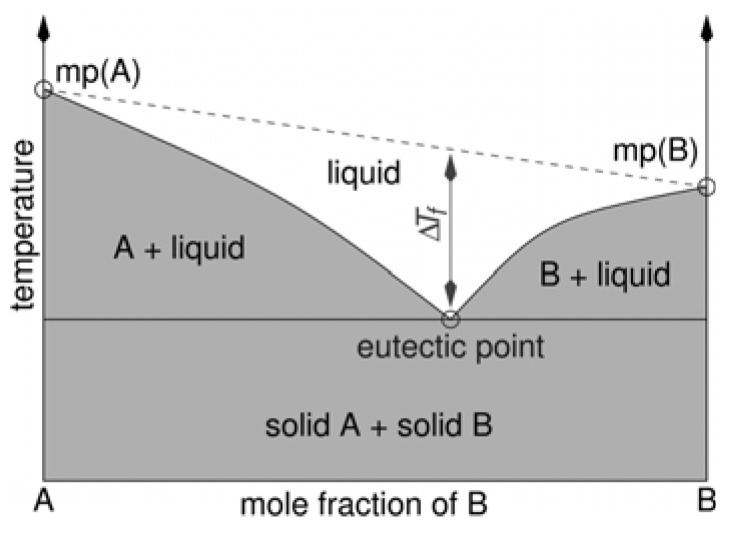
Schematic representation of the eutectic point in a binary (A + B) phase diagram [16].

**Figure 3 materials-16-00627-f003:**
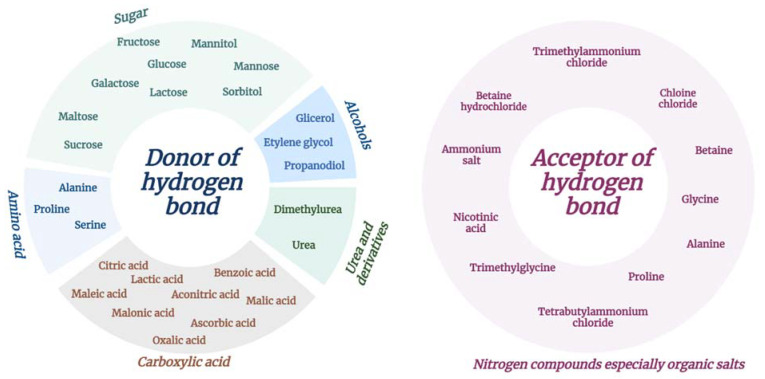
Components of natural deep eutectic solvents divided into hydrogen bond acceptors and hydrogen bond donors) (created with BioRender.com).

**Figure 4 materials-16-00627-f004:**
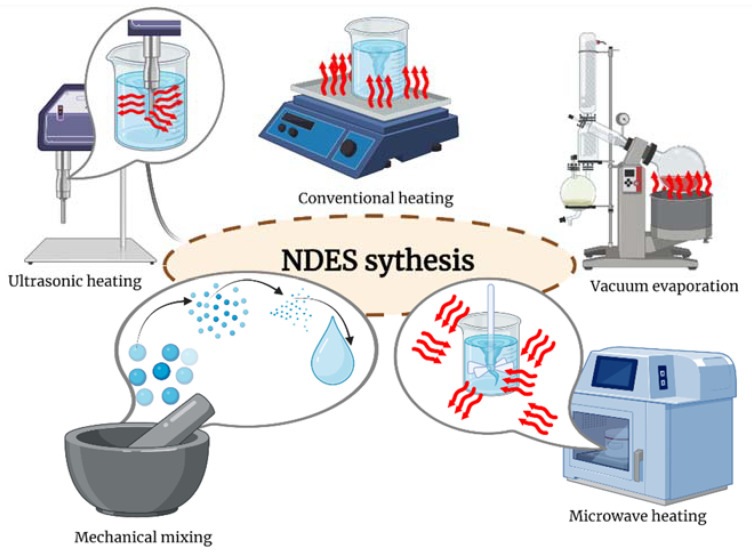
Methods for selecting the heating sources of reactants for obtaining NDES (created with BioRender.com).

**Figure 5 materials-16-00627-f005:**
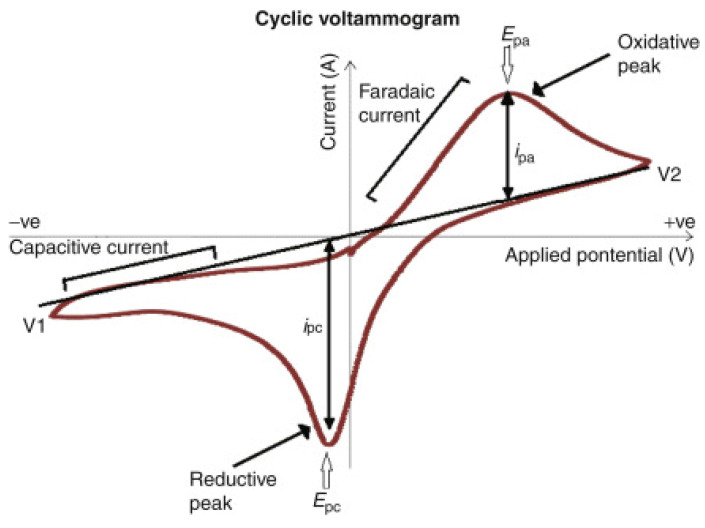
A schematic diagram of a cyclic voltammogram according to the IUPAC convention: peak cathodic potential (E_pc_), peak anodic potential (E_pa_), difference between the cathodic current and the resting current (i_pc_) and difference between the anodic current and the resting current (i_pa_) [52].

**Figure 6 materials-16-00627-f006:**
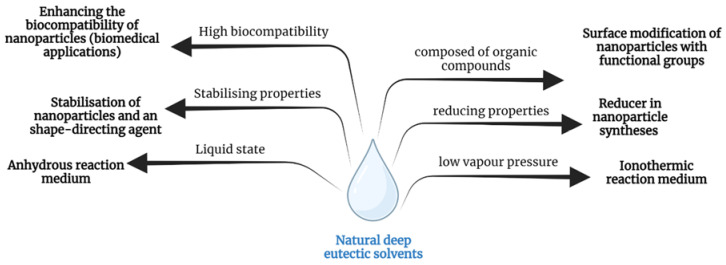
Properties and applications of natural deep eutectic solvents in nanotechnology (created with BioRender.com).

**Table 1 materials-16-00627-t001:** Classification and examples of deep eutectic solvents.

Type	Composition
Type I	Organic salt + metal chloride (e.g., choline chloride–${\mathrm{ZnCl}}_{2}$)
Type II	Organic salt + metal chloride hydrate (e.g., choline chloride–${\mathrm{CoCl}}_{2}\cdot 6H_{2}O$)
Type III	Organic salt + hydrogen bond donor (e.g., choline chloride–urea)
Type IV	Hydrogen bond donor + metal chloride hydrate (e.g., urea–${\mathrm{ZnCl}}_{2}$)

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
