# Peer review of "Natural Deep Eutectic Solvents in the Synthesis of Inorganic Nanoparticles"

_materials, 2023, doi:10.3390/ma16020627_

Round 1

Reviewer 1 Report

The reviewed work is in a new field and the summary of results and papers presented will help other researchers. The work is interesting and I believe it should be published.

I include my comments below:

1. Abstract- after reading the abstract, I could not tell exactly what the paper would be about and I think the author could have emphasised this more clearly. I suggest a figure should be placed below the abstract.

The last paragraph is completely unclear.

2. the definition of NDES is not clear to me, maybe the author would try to put some kind of a diagram to make it easier to understand.

3. the chapter on synthesis is interesting but I miss the introduction of why it is in this paper.

4. the Properties of NDES chapter perhaps should be called Properties and characterisation of NDES?

4.1 Chapter - Theory of holes in my opinion needs a few sentences explanation due to the nature of the publication for a wider audience.

4.2 chapter - sentence: The study of metal ion speciation .... to the end of the paragraph is unclear.

4.4 chapter- The explanation of the CV technique seems very confusing to me, could the author please rewrite this description?

Can the author add a short speculative chapter (to emphasise it more) the use of NDES in other scientific field.

Author Response

Dear Reviewer,

Many thanks for the submitted review. In the text, changes have been highlighted in blue. Only minor grammatical, linguistic and punctuation corrections have not been shown.

I believe that this manuscript will be suitable for publication in the journal Materials. Thank you very much for your attention and we look forward to receiving your feedback.

  1. Comments and Suggestions for Authors

The reviewed work is in a new field and the summary of results and papers presented will help other researchers. The work is interesting and I believe it should be published.

I include my comments below:

  1. Abstract- after reading the abstract, I could not tell exactly what the paper would be about and I think the author could have emphasised this more clearly. I suggest a figure should be placed below the abstract.

Answer: The abstract has been improved and the figure has been moved near the abstract.

The last paragraph is completely unclear.

Answer: According to the suggestion the paragraph has been revised and improved. The chapter heading has also been changed to make it more readable

  1. The definition of NDES is not clear to me, maybe the author would try to put some kind of a diagram to make it easier to understand.

Answer: The definition of the term NDES has been improved (Paragraph 2.3).

  1. the chapter on synthesis is interesting but I miss the introduction of why it is in this paper.

Answer. I agree with the suggestion as much as possible. A paragraph has been added to the chapter to explain the purpose of describing this issue in the article, making the text clearer.

  1. the Properties of NDES chapter perhaps should be called Properties and characterisation of NDES?

Answer: According to the comment, the title of the paragraph has been changed.

4.1 Chapter - Theory of holes in my opinion needs a few sentences explanation due to the nature of the publication for a wider audience.

Answer: Thank you for your comment. The paragraph has been corrected and expanded to make it understandable to readers (Paragraph 4.1).

4.2 chapter - sentence: The study of metal ion speciation .... to the end of the paragraph is unclear.

Answer: Text has been corrected.

4.4 chapter- The explanation of the CV technique seems very confusing to me, could the author please rewrite this description?

Answer: The paragraph 4.4 has been improved. The header has also been changed.

Can the author add a short speculative chapter (to emphasise it more) the use of NDES in other scientific field.

Answer: In the text, the main areas in which NDESs are used have been added. However, the paragraph with applications have not been described separately (paragraph 5), due to the number of branches that would need to be described, which could make the text difficult to read.

Reviewer 2 Report

The Manuscript entitled “Natural deep eutectic solvents in the synthesis of inorganic nanoparticles” contributes significantly to the field. However, the Manuscript is not well organised. The Manuscript can be considered for publication after the major revision. Please revise the Manuscript addressing the following comments in the revised version. 

·         novelty of this Manuscript? Add a sentence in the abstract to show the novelty and need of this review paper. 

·         The authors have not mentioned the importance of NDES and EDS in nanotechnology. Please give historical background on the introduction of EDS and NEDS, including how this contributed to filling the research gap. 

·         “Among the most obvious and best described are the use of NDEs in electronics, in medicine and in biochemical processes [4]”. What is the difference between NDES and NDEs?

·         There exists an inconstancy between the abbreviations and their expanded forms. All the abbreviations need to be expanded for the first appearance in the Manuscript. Later do not need to repeat the expanded form again and again. Check all such mistakes and revise them. 

·         Repetition of paragraph (two times) on page 3. The sentence beginning, “Deep eutectic solvents are divided into several types [14]”. In addition, both the same paragraph has different citation numbers.

·         “Tab. 1. Classification and examples of deep eutectic solvents” Check the spelling.

·         Unnecessary use of bold characters (Fig 1)

·         Tab. 4. Please follow single patterns for all tables.

·         The NDES comparison with DES and IL is rarely pronounced throughout the review. It is suggested to add strong points to classify based on advantages/disadvantages (mainly from heading 5. Natural deep eutectic solvents applications in nanotechnology).

·         Many typos, grammatical errors and unnecessary punctuation have been noticed throughout the Manuscript. The authors need to proofread the Manuscript.

Author Response

Dear Reviewer,

Many thanks for the submitted review. In the text, changes have been highlighted in blue. Only minor grammatical, linguistic and punctuation corrections have not been shown.

  1. Comments and Suggestions for Authors

The Manuscript entitled “Natural deep eutectic solvents in the synthesis of inorganic nanoparticles” contributes significantly to the field. However, the Manuscript is not well organised. The Manuscript can be considered for publication after the major revision. Please revise the Manuscript addressing the following comments in the revised version. 

  1. novelty of this Manuscript? Add a sentence in the abstract to show the novelty and need of this review paper. 

Answer: The abstract has been improved. The novelty has been highlighted.

  1. The authors have not mentioned the importance of NDES and EDS in nanotechnology. Please give historical background on the introduction of EDS and NEDS, including how this contributed to filling the research gap. 

Answer: In the introduction, the historical background is explained. The types of DES and the differences between them are compared and described.

  1. “Among the most obvious and best described are the use of NDEs in electronics, in medicine and in biochemical processes [4]”.What is the difference between NDES and NDEs?

Answer: Thank you for your attention. An error has occurred in the text and should be NDES. The entire text has also been checked to eliminate similar errors.

  1. There exists an inconstancy between the abbreviations and their expanded forms. All the abbreviations need to be expanded for the first appearance in the Manuscript. Later do not need to repeat the expanded form again and again. Check all such mistakes and revise them. 

Answer: As suggested, the text has been corrected and unnecessary translations removed.

  1. Repetition of paragraph (two times) on page 3. The sentence beginning, “Deep eutectic solvents are divided into several types [14]”. In addition, both the same paragraph has different citation numbers.

Answer: Thank you for your comment.  The paragraphs have been corrected and merged into 1.

  1. “Tab. 1. Classification and examples of deep eutectic solvents”Check the spelling.

Answer: The table and captions have been corrected.

  1. Unnecessary use of bold characters (Fig 1)

Answer: The figure has been corrected as suggested.

  1. 4.Please follow single patterns for all tables.

Answer: Table and symbols have been improved, as suggested.

  1. The NDES comparison with DES and IL is rarely pronounced throughout the review. It is suggested to add strong points to classify based on advantages/disadvantages (mainly from heading 5. Natural deep eutectic solvents applications in nanotechnology).

Answer: Sections 1 and 2 additionally explain the advantages of NDES over DES and IL and give reasons why they are used in nanotechnology as opposed to other solvents.

  1. Many typos, grammatical errors and unnecessary punctuation have been noticed throughout the Manuscript. The authors need to proofread the Manuscript.

Answer: The article has been linguistically corrected, the text has been read and revised again.

Reviewer 3 Report

The author reviewed the composites, synthesis, properties and the applications of natural deep eutectic solvents for the synthesis of nanoparticles. In my opinion, the topic is interesting to the community. However, the current version has some crucial problems, and therefore, it needs significant improvements before publication.

The first problem is that the manuscript does not have a section of conclusion, which resulted in an incomplete manuscript.

The second problem is the lack of theoretical backgrounds. The authors did not explain the theories behind the properties, which led to the difficulty in understanding the manuscript. Here are some exemplar points:

1.     Page 2: When explain the eutectic point, the readers would like to see a phase diagram and the introduction of all the terms.

2.     Page 7: The authors should briefly describe the hole theory. It is very inconvenient for the readers that the authors only referred to Ref [6]. They should also list and briefly explain the Stockes-Einstein, Arrhenius and VFT equations for the dynamic properties so that the discussion about the effect of temperature would be meaningful.

3.     Section 4.3 on page 8: In the lines of “In determining the size of nanoparticles, the DLS method is based on determining the changes in the reflection of light by moving nanoparticles”, did the author want to describe dynamic light scattering? What is the full name of DLS? The principle of DLS is not reflection. The author should recheck the definition of DLS.

4.     Section 4.4 on page 8: When describing the cyclic voltammetry, the authors mentioned: “These graphs represent a reversible redox process, where two peaks are depicted.” They should provide a graph as an example and explain the shape of the curve with the redox reaction of NDES. The author should also confirm if the redox involves proton transfer along with the electron transfer.

The third problem is the language and organization. Some sentences are not scientific, and some duplicated paragraphs should be combined.

1.     Page 1: ‘are the use of NDEs in electronics,’, ‘NDEs’ should be ‘NDES’.

2.     Page 3: The second and third paragraphs have the duplicated contents describing the four types.

3.     Section 4.2 on page 7. The sentence is hard to understand and the author need to rephrase it: “One of the key questions when considering the reactions occurring between metal forms and DES is the forms that form during the bonding processes.”

4.     The references are not consistent or correct:

a.     Ref 18: doi directs to the supplementary file.

b.     Section 5.1: “Their main advantage is their liquid form without the use of water or simple organic solvents (Abo-Hamad et al. 2015a; Tomé et al. 2018; Gajardo-Parra et al. 2019; Shafie et al. 2019).”

c.      Section 6 on page 16: In “According to Hansen et al. this can be achieved through parallel efforts including:”, no reference is provided for Hansen et al.

5.     Section 5.1: “Reaction media for the synthesis of nanomaterials” and Section 5.4: “Examples of preparation of nanoparticles using NDES”. What are the differences? Why are they the separated sections?

Author Response

Dear Reviewer,

Many thanks for the submitted reviews. All the comments have been taken into account. In the text, changes have been highlighted in blue. Only minor grammatical, linguistic and punctuation corrections have not been shown.

Due to both reviews being received simultaneously in one message, the first review addressed all comments. I believe that this manuscript will be suitable for publication in the journal Materials. Thank you very much for your attention and we look forward to receiving your feedback.

Comments and Suggestions for Authors

The author reviewed the composites, synthesis, properties and the applications of natural deep eutectic solvents for the synthesis of nanoparticles. In my opinion, the topic is interesting to the community. However, the current version has some crucial problems, and therefore, it needs significant improvements before publication.

The first problem is that the manuscript does not have a section of conclusion, which resulted in an incomplete manuscript.

Answer: In the first version, it was considered that this form would suit the readers, but as suggested, the structure of the article has been changed and a conclusion section has been added.

The second problem is the lack of theoretical backgrounds. The authors did not explain the theories behind the properties, which led to the difficulty in understanding the manuscript. Here are some exemplar points:

  1. Page 2: When explain the eutectic point, the readers would like to see a phase diagram and the introduction of all the terms.

Answer: As suggested, a schematic representation of the eutectic point in a binary (A+B) phase diagram has been added.

  1. Page 7: The authors should briefly describe the hole theory. It is very inconvenient for the readers that the authors only referred to Ref [6]. They should also list and briefly explain the Stockes-Einstein, Arrhenius and VFT equations for the dynamic properties so that the discussion about the effect of temperature would be meaningful.

Answer: Due to the length of the article and the focus on NDES applications in nanotechnology, only the main points important for NDES are mentioned. However, this section has been expanded and descriptions have been added.

  1. Section 4.3 on page 8: In the lines of “In determining the size of nanoparticles, the DLS method is based on determining the changes in the reflection of light by moving nanoparticles”, did the author want to describe dynamic light scattering? What is the full name of DLS? The principle of DLS is not reflection. The author should recheck the definition of DLS.

Answer: Kindly forgive the abbreviations used and the absence of the full name of the method. The paragraph has been corrected to make it readable.

  1. Section 4.4 on page 8: When describing the cyclic voltammetry, the authors mentioned: “These graphs represent a reversible redox process, where two peaks are depicted.” They should provide a graph as an example and explain the shape of the curve with the redox reaction of NDES. The author should also confirm if the redox involves proton transfer along with the electron transfer.

Answer: The paragraph on the cyclic voltammetry has been corrected and a diagram has been added.

  1. The third problem is the language and organization. Some sentences are not scientific, and some duplicated paragraphs should be combined.

Answer: Article has been reread and text has been improved.

  1. Page 1: ‘are the use of NDEs in electronics,’, ‘NDEs’ should be ‘NDES’.

Answer: Similar errors have been corrected and the entire text read again to eliminate other errors.

  1. Page 3: The second and third paragraphs have the duplicated contents describing the four types.

Answer: The text has been improved.

  1. Section 4.2 on page 7. The sentence is hard to understand and the author need to rephrase it: “One of the key questions when considering the reactions occurring between metal forms and DES is the forms that form during the bonding processes.”

Answer: The text has been improved.

  1. The references are not consistent or correct:
  • Ref 18: doi directs to the supplementary file.
  • Section 5.1: “Their main advantage is their liquid form without the use of water or simple organic solvents (Abo-Hamad et al. 2015a; Tomé et al. 2018; Gajardo-Parra et al. 2019; Shafie et al. 2019).”
  • Section 6 on page 16: In “According to Hansen et al. this can be achieved through parallel efforts including:”, no reference is provided for Hansen et al.

Answer: Literature has been revised and the whole reference has been checked.

  1. Section 5.1: “Reaction media for the synthesis of nanomaterials” and Section 5.4: “Examples of preparation of nanoparticles using NDES”. What are the differences? Why are they the separated sections?

Answer: In paragraph 5.1, the author focused on the use of NDES as a solvent to carry out reactions to obtain nanoparticles. Paragraph 5.4 already reported specific examples of the use of NDES in nanoparticle synthesis, including summarised in tables examples of applications. 

Round 2

Reviewer 2 Report

The authors made relevant modifications during the revision. The manuscript can be considered for publication.

Author Response

Thank you very much for your help in improving the article. 

Reviewer 3 Report

The author has improved the manuscript but some problems still exist:

The first problem is that the manuscript does not have a section of conclusion. The readers would like to see a complete paper with the summary and conclusions at the end.

The second problem is the lack of theoretical backgrounds. The authors did not explain the theories behind the properties, which led to the difficulty in understanding the manuscript. Here are the  points:

1.  Section 2.1 on page 3: When explain the eutectic point, the readers would like to see a phase diagram and the introduction of all the terms.

2.    Section 4.1 on page 8: The authors should briefly describe the definition of hole theory. It is inconvenient to look for the definition in Ref [16]. Moreover,  Stockes-Einstein, Arrhenius and VFT equations should also be explained for the dynamic properties. Does NDES have the glass transition since VFT was used?

3.     Section 4.3 on page 9: In the lines of “In determining the size of nanoparticles, the DLS method is based on determining the changes in the reflection of light by moving nanoparticles”, did the author want to describe dynamic light scattering? What is the full name of DLS? The principle of DLS is not reflection. The author should recheck the definition of DLS.

4. Section 4.4 on page 10: Does the redox involve proton transfer? This should be important for NEDS to serve as catalyst.

In addition to the above, I can not find why Section 5.1: “Reaction media for the synthesis of nanomaterials” and Section 5.4: “Examples of preparation of nanoparticles using NDES” are separated. What are the differences? 

Author Response

Due to both reviews being received simultaneously in one message, the first review addressed all comments.